# Equivalent Circuits for Microwave Metamaterial Planar Components

**DOI:** 10.3390/s24072212

**Published:** 2024-03-29

**Authors:** Romolo Marcelli

**Affiliations:** CNR-IMM Roma, Via del Fosso del Cavaliere 100, 00133 Roma, Italy; romolo.marcelli@cnr.it; Tel.: +39-06-4548-8536

**Keywords:** microwaves, metamaterials, equivalent circuits

## Abstract

Metamaterial components and antennas are based on the general understanding that an artificial structure composed of adequately designed and manufactured elementary cells or arrays has unusual resonance and propagation properties. Metamaterials exhibit equivalent values of the dielectric constant and magnetic permeability that are both negative simultaneously, in contrast with ordinary materials. Single elements, periodic, or quasi-periodic configurations can be suitable for a metamaterial response. In this paper, equivalent circuits for microwave propagation and resonance are compared, deriving a lumped element modeling complementary to those already available in the literature, with a particular focus on planar resonating devices and calculating the effective value for the dielectric constant and the magnetic permeability directly from experimental findings using the impedance (*Z*-parameters) notation.

## 1. Introduction

Metamaterial components and antennas for high-frequency applications have been widely studied during the last two decades for their unusual properties suitable for novel applications, and several papers and books are available on this topic. See, for instance, [1,2,3,4]. As is well known, the seminal theoretical paper from Veselago inspired this topic [5], and when the technology was mature enough to create structures suitable for the predicted properties of the dielectric constant and the magnetic permeability, a significant increase in scientific contributions and potential applications happened. The main characteristic of the special materials introduced by [5] was the simultaneous negative sign of the dielectric constant and magnetic permeability, originating the excitation of backward waves in the propagation medium. The frequency dispersion is a fundamental requirement to accept a negative sign for both quantities, while still assuring a positive value for the involved energy. In this way, energy flows away from the source, but the wavefront is coming back in the opposite direction. That scientific contribution was followed by experimental attempts to demonstrate such a prediction with media and devices properly designed and excited.

Controlling wave propagation in the forward or backward direction allows for additional telecommunication solutions and was considered an advantage for controlling the beam direction of an antenna. Focusing on a beam was also an exciting result for [5], where it is discussed how a negative refractive index modifies the wave path. This discussion was followed by additional theoretical considerations in [6] to predict the possibility of focusing a wave with a flat lens for better imaging resolution. Experimental proof and the practical application of such a lens were first demonstrated in [7]. Any frequency range can benefit from the above-cited effects due to the simultaneously negative value of the dielectric constant and magnetic permeability, and planar components for microwave signal processing based on the metamaterial solution can be improved compared with the classical ones. Looking at the dispersion characteristics offered by metamaterials, additional frequencies can be excited, and zeroth order resonators can be designed. The last concept is helpful in exciting a zero-value of the wavevector even at a non-zero frequency, and it was also applied to the miniaturization of microwave patch antennas [8,9].

The circuital modeling of metamaterial devices for high-frequency components and antennas was studied initially to identify the origin of the difference with respect to traditional configurations. It has to be stressed that circuits are familiar to researchers dealing with devices and applied electromagnetics. At the same time, scientists working on material science are usually interested in something other than an equivalent circuit; in propagation characteristics, for example, they are interested in electromagnetic materials. Moreover, an equivalent circuit is only sometimes possible for high frequencies because a distributed structure is rigorously modeled when additional components, namely parasitic contributions, are present. On the other hand, for many configurations and sometimes also for heuristic purposes, the lumped element modeling can be considered valid up to the beginning of the millimeter wave range, i.e., for frequencies not higher than 30 GHz. Most initial papers focused immediately on the lumped inductive or capacitive contributions of specific circuit branches, correlating them to their classical or novel responses, utilizing a transmission line (TL) approach [10]. To model simple TLs, we have two possibilities, neglecting losses and parasitic components and using the so-called T-model: (i) an inductor in line with a shunt capacitor and (ii) a capacitor in line with a shunt inductor. A classical T-model for the TLs is represented in Figure 1, comparing in (a) and (b) the configurations used to model an ordinary TL and a metamaterial structure, respectively.

The change in the microwave response obtained by inverting the position of the inductor and the capacitor has been discussed in several papers and books, mainly to derive the values of the equivalent dielectric constant and the equivalent permeability of the metamaterial configuration considered as a “material” where the wave passes through, like in [10]. Sometimes, only inductors and capacitors are used, to simplify the approach or for educational purposes, which are enough to obtain the expected response of metamaterial structures. In this ideal case, only the so-called right-handed (RH) and left-handed (LH) TLs are considered, with consequent approximations for the definition of cut-off frequencies, the magnetic permeability values, and the dielectric constant. 

Nevertheless, in real cases, additional resistors and conductances must be introduced, as well as parasitic capacitances and inductors. For this reason, it is correct to have a more complete view of the problem, defining an elementary cell containing all the contributions mentioned above, leading to a composite-right-left-handed (CRLH) TL [10]. This circuit implementation is advantageous for having a balanced configuration without forbidden gaps in the dispersion relation. A careful design of propagating and resonating metamaterial devices might account for a balanced CRLH behavior because the unbalanced circuit has intrinsically a narrow-band response. From the analysis of the impedances, it turns out that, in principle, the balanced configuration is electrically matched for each frequency, leading to a broader band of operation.

On the other hand, depending on the dominant response in the TL, we can always say that a branch can be mainly inductive or capacitive, using the Smith Chart’s canonical definitions, available, for instance, in [11,12], and fit the equivalent circuit using experimental data.

Based on this approach, simplified modeling will be discussed to determine the kind of lumped components necessary to fully describe the response of the measured TL or TL-based configuration. Similarly, filters could be studied to extract their general response as ordinary or metamaterial structures. In this paper, we shall develop a simple criterion based on analyzing the experimentally measured impedance (*Z*), using the definition of the *Z*-parameters and their application in interpreting the experimental data in actual devices. Such analysis aims to define a criterion for designing metamaterial building blocks suitable for resonant arrays and to verify their metamaterial response experimentally. Moreover, the study of configurations inspired by metamaterial geometries rarely concludes by confirming that the structure as a whole behaves like a metamaterial component or antenna by looking at the equivalent dielectric constant and magnetic permeability. For this reason, this contribution is intended to propose an experimental method to determine the RH (ordinary) or LH (metamaterial) nature of a microwave planar component focusing on the entire configuration, including implicitly parasitic lumped components and feeding networks.

## 2. General Criteria to Determine the Metamaterial Nature of a Transmission Line

A linear, symmetric, and reciprocal basic transmission line (TL) is defined in [11] and other classical books like [12]. T-sections or Π-sections are introduced to describe the elementary cell, using impedances to explicitly translate into inductive or capacitive contributions. The most general structure that can be plotted using this approach for a T-modeled elementary cell is shown in Figure 2.

The structure in Figure 2b is the most straightforward configuration for a TL section. Still, it is also very general because, except for magnetic devices, most passive signal processing configurations, like resonators, filters, delay lines, or phase shifters, are electrically matched with the same input/output impedance.

A microwave device’s frequency response is typically achieved by measuring the so-called scattering (S) parameters, i.e., non-dimensional complex quantities computed as the ratio between input and output signals at the ports of a vector network analyzer (VNA). A VNA is an instrument that records the amplitude and phase of signals reflected from a port or passing through the device, and multi-port configurations are commercially available. The S-parameters are defined by a matrix with rows and columns whose number depends on the number of ports of the device under test (DUT). Since the DUT can be modeled in terms of impedances, an analytical approach is available to transform the S-matrix, with elements $S_{ij}$ and i,j=1,2 into the Z-matrix, with elements $Z_{ij}$. How to pass from *S*- to *Z*-parameters is well documented in [11,12]. Other approaches are also available based directly on the *S*-parameters [13].

Once we have derived the Z-parameters from post-processing the measurements, we can use the equations governing the response of the DUT in the following way:(1)$$\left\{\begin{array}{c} Z_{11}=Z_{1}+Z_{2}\\ Z_{12}=Z_{21}=Z_{2}\\ Z_{22}=Z_{1}+Z_{2}=Z_{11} \end{array}\right.$$

As is well-established in microwave engineering design, the impedance analysis can be performed using the Smith Chart for an immediate visual check of the dominant contribution of the equivalent lumped component. This approach can be easily applied to the experimental data, and the qualitative result of the analysis is summarized in the following plot, where the two fundamental complex components to be considered will be $Z_{1}=Z_{11}-Z_{21}$ and ${Z_{2}=Z}_{21}$, and the imaginary part will give evidence of the main contribution in the response of the individual branches of the T-section. In practice, using a reflection and a transmission measurement for a symmetric and reciprocal TL will help to describe the two-port network’s complete response. This result is immediate for a lossless section of a TL, where all the possible responses lead to the plot in Figure 3.

The interpretation of Figure 3 is quite simple for the ideal, lossless case because there is no additional contribution to explain the measured response. It might be helpful to remember that the plot in Figure 3 is directly inspired by contributions like [1,10], where the diagram was proposed in terms of the sign for the dielectric constant and the magnetic permeability.

When ${Im(Z}_{21})=Im(Z_{2})>0$ (inductive behavior), we can have ${Im(Z}_{1})>0$ if an inductor is present (equivalent circuit in the upper right corner); otherwise, $Im(Z_{1})<0$ will correspond to a capacitor (upper left corner). When $Im(Z_{21})<0$, then $Im(Z_{1})$ can be again negative or positive, leading to the lower left corner circuit or the lower right corner one, respectively. Despite the lack of generality because losses and parasitic contributions are not included, this criterion is accurate enough to distinguish among possible responses of the elementary cell, eventually summarizing the total DUT behavior in the case of two-port resonating structures. We must remember that an equivalent circuit is always an approximate view of a more complex structure, and sometimes, the most appropriate circuit fitting the frequency response of the DUT is not intuitive, especially at high frequencies, where distributed contributions have to be considered. In the presence of losses, the approach maintains its generality because its imaginary part can formally substitute the above-introduced impedances, and negative or positive values of the reactive contribution can be conveniently observed using a Smith Chart representation. In formulae, for a metamaterial TL (i.e., the LH structure in the upper left corner of Figure 3) without losses, we can write the following:(2)$$Z_{21}=Z_{2}>0\;Z_{2}=i\omega L\;\alpha \;\mu_{r}\;Z_{1}<0\;Z_{1}=-\frac{i}{\omega C}\alpha \frac{1}{\varepsilon_{r}}$$

So far:(3)$$\mu_{r}\;\alpha \;Im\left\{Z_{2}\right\}=Im\left\{Z_{21}\right\}$$
(4)$$\varepsilon_{r}\;\alpha \frac{1}{Im\left\{Z_{1}\right\}}=\frac{1}{Im\left\{Z_{11}-Z_{21}\right\}}$$

When both quantities on the right hand of the above equations, experimentally derived, are negative, we can define the TL circuit as a metamaterial structure, i.e., for a capacitive input branch and an inductive shunt branch in the T-equivalent circuit [10]. The generality of this method is based on the experimental derivation of the *Z*-parameters from the *S*-parameters, and the model can include the I/O lines bringing the signal to a filter or other resonant structures. The more general approach is based on Figure 2b, where a simple T-model is diagrammed, and the *Z*-circuital elements are linked to the *Z*-parameters obtained using the *S*-parameters. Of course, as discussed in the literature, the model is phenomenological and convenient because it allows the utilization of circuital concepts translated into material characteristics. Since we are not interested in a very detailed description of the circuit but in recognizing its response as an ordinary structure or a metamaterial one, any other helpful parameter to have an exact quantitative full description of the circuit can be neglected in this heuristic analysis. In conclusion, to have a TL that can be represented by a right-handed (RH) ordinary circuit or a left-handed (LH) metamaterial one in a two-port symmetric and reciprocal network is demanded for the analysis of the *Z*-parameters, and the sign of $Im\left\{Z_{21}\right\}$ and $Im\left\{Z_{11}-Z_{21}\right\}$.

It might be stressed that a metamaterial structure can have additional properties, and evaluating the sign of the equivalent dielectric constant and magnetic permeability linked to the lumped components used to model the configuration is not enough to fully understand its response and novelty with respect to an “ordinary material”. On the other hand, we are now focused on this possible characteristic of the circuit, avoiding a complete discussion implying those other non-ordinary responses, like backward radiation in the case of antennas or phase shifters.

## 3. Metamaterial Resonators

Two main advantages are recognized in the scientific literature about metamaterial resonators: (1) a sharp response at the resonance frequency, contributing to a higher quality factor, and (2) a reduction of the electrical size, using a physical size smaller than the wavelength of the incident wave [13]. In our case, an additional advantage is given by the fixed footprint, and the internal complexity has the task of changing the operation frequency. The last characteristic makes the Sierpinski triangles suitable for realizing arrays with the fine-tuning of bandwidth and frequency of operation, combining different geometries within a purposely designed array. In our study, we propose the utilization of equilateral triangles because they can be easily integrated into a more complicated array, but this is a possibility and not a limitation.

The general configuration in Figure 2b can be used for modeling circuits with $Z_{2}$ describing a resonant circuit. Alternatively, we can include a resonant circuit within two ordinary transmission line sections to bring the signal to the resonator. Using $Z_{2}$ as a resonator, we can have, for a band-stop configuration, a series of $L_{r}$ and $C_{r}$ fed by a TL with characteristic impedance $Z_{0}$ leading to the classical equations:(5)$$\omega_{r}=\frac{1}{\sqrt{L_{r}C_{r}}}Z_{0}=\sqrt{\frac{L_{r}}{C_{r}}}$$

From the above equations, we can obtain:(6)$$L_{r}=\frac{Z_{0}}{\omega_{r}}\;C_{r}=\frac{1}{Z_{0}\omega_{r}}$$

So, from the experimental determination of the characteristic impedance (or its definition by design) and the frequency of resonance $f_{r}=\frac{\omega_{r}}{2\pi }$, we obtain the values of the equivalent lumped components for the resonator. The measurement of the quality factor leads to the additional element, i.e., the resistor, which describes the losses of the resonator. For a series resonator:(7)$$Q=\frac{\omega_{r}L_{r}}{R_{r}}=\frac{1}{\omega_{r}C_{r}R_{r}}=\frac{\omega_{r}}{{∆\omega }_{r}}$$
where $R_{r}$ is the resonator resistance, and Q is the quality factor obtained measuring the frequency of resonance and the linewidth ${∆\omega }_{r}$ of the resonator, from which we can get the resistance. 

We can also write, in general, that $L_{r}=L_{r0}\mu_{r}$ and $C_{r}=C_{r0}\varepsilon_{r}$ where the suffix “0” is valid for the in-vacuum equivalent lumped component, and the dielectric constant and magnetic permeability as a multiplication factor are included when an actual medium with its properties is considered. The above concepts are essential for a complete numerical determination of the lumped equivalent circuital components because both inductance and capacitance can be regarded as proportional to the magnetic permeability $\mu_{r}$ and dielectric permittivity $\varepsilon_{r}$, respectively. On the other hand, when the circuital analysis aims to determine a negative sign for both quantities, i.e., $\mu_{r}$ and $\varepsilon_{r}$, the proportionality is enough because $L_{r0}$ and $C_{r0}$ always have positive values.

Nevertheless, a resonator is characterized by vanishing the reactive contribution at the resonance frequency, maintaining only the resistive part, i.e., the real part of the impedance. For this reason, it is not fully correct to determine a possible metamaterial response at resonance with this approach because, in this case, $Im\left\{\left.Z_{2}\right\}\right.=0$, and the initial equations must be modified to obtain reasonable values accounting for a possible reactive contribution. Since TLs feed a resonator, we can consider the entire device, including a cascaded configuration where the resonator is placed in the middle of the structure with two TLs at the input/output ports. Following this approach, the schematic diagram of the entire network could be represented in Figure 4.

Then, a two-step measurement (or theoretical evaluation) can be performed: (1) determine the I/O impedances of an ordinary TL and derive the *Z*_1_ and *Z*_2_ values corresponding mainly to an inductor and a capacitor, respectively; (2) determine *Z_r_* moving the I/O ports to the internal ones with a de-embedding procedure. On the other hand, a simplified approach can be followed, considering that the entire structure could be suitable for an ordinary or metamaterial response. We can define a ${Z_{2}}^{\prime }$ component, including part of the feeding system and the resonator itself. In this case, we can simplify the circuit in Figure 4 using the equivalent one presented in Figure 5.

So far, the initial equations linking the *Z*-parameters to the impedances of the equivalent circuit can be re-written as follows:(8)$$\left\{\begin{array}{c} Z_{11}=Z_{1}+{Z_{2}}^{\prime }\\ Z_{12}=Z_{21}={Z_{2}}^{\prime }\\ Z_{22}=Z_{1}+{Z_{2}}^{\prime }=Z_{11} \end{array}\right.$$

Currently, ${Z_{2}}^{\prime }$ is a more complicated network, but it still preserves the meaning of an equivalent impedance that helps determine the ordinary or metamaterial response of the entire configuration.

At the end, it is as follows:(9)$$Z_{1}=Z_{11}-Z_{21}\;\mathrm{and}\;{Z_{2}}^{\prime }=Z_{21}$$

It means that having a metamaterial component is equivalent to fulfilling the following updated equations:(10)$$Im\left(Z_{2}^{\prime }\right)={Im(Z}_{21})>0\;{Im(Z}_{1})=Im(Z_{11}-Z_{21})<0$$

## 4. Experimental Determination of the Metamaterial Nature of a Resonator

The scientific literature has characterized several resonator structures, claiming their metamaterial nature. Most efforts have been dedicated to the so-called split ring resonator (SRR) [13] and recent publications have been focused on a lossless network composed of input/output lines and the resonator itself [14]. On the other hand, independently of the loss mechanisms and their inclusion in the model, an experimental approach can help in revealing the intrinsic nature of the resonator considered with all the details, including losses and feeding network. In this section, we prefer to use the definition of metamaterial-inspired resonators, checking the possible metamaterial response according to the simple method developed in the past paragraphs. A similar approach has been utilized in [15] without entering the details of the analytical derivation. A complete theoretical treatment has been proposed in [16,17,18,19,20,21,22,23] to cite a few of the available papers in the literature, with the ambition to evaluate analytically all the possible contributions to be included, assisted by software simulation and experimental findings to validate the model. The chapter contribution in [22] perfectly summarizes the specific advantages of antennas suitable for forward and backward propagation and multi-band response, while in [23,24], an explicit derivation of the dielectric constant and magnetic permeability is also given, using the *S*-parameters. In [24], the original approach proposed and developed to obtain the material parameters using free wave propagation is utilized [25,26,27,28,29]. In this paper, and specifically in this section, we shall present experimental plots related to the quantities defined at the end of the previous paragraph, giving evidence for the frequencies where it is reasonable to assume a metamaterial response. The advantage of the proposed phenomenological approach is that it utilizes *Z*-parameters as experimental findings of the studied structure. This method gives immediate feedback about the possible metamaterial nature of the device without introducing a very detailed description. In fact, a deeper analytical derivation could fail because of the impossibility of modeling exactly the lumped elements belonging to a CRLH configuration, including parasitic components, as a function of frequency.

In [14], we analyzed the electrical response of two resonators based on (1) a fractal-inspired geometry with the shape of Sierpinski triangles and (2) simple U-shapes. Intuitively, a Sierpinski geometry is expected to show a metamaterial behavior, but we have preliminarily seen that the U-shape is also favored when looking for a metamaterial structure.

One consequence of the metamaterial design studied in the literature is the recognition that this characteristic is fulfilled only in a narrow bandwidth, i.e., to have a negative value for permeability and permittivity or other properties like forward and backward wave onset.

The device we shall analyze is a triangular resonator made by a full metal patch coupled to a line embedded in a coplanar waveguide (CPW) configuration compared with the same structure hosting a Sierpinski triangle with a first-level internal complexity, which can be considered as a fractal geometry, often viewed as a relative of the metamaterial geometries. Analogously to [14], we define them as C0 and C1, respectively. The two configurations are shown in Figure 6. The Sierpinski geometry has been studied for decades from a mathematical point of view, and it has been proposed especially in high-frequency antennas for multi-band purposes [30]. In this paper, owing to the possibility of having a fixed footprint of the resonator, the frequency of operation is tuned to change the internal complexity.

The resonator size has been imposed according to the resonance frequency expected for an equilateral triangle, as a first approximation, which depends on the edge length. As demonstrated in [31], the coplanar excitation is more difficult to be understood from the spectral point of view compared with a microstrip excitation, but the frequency of resonance of the main mode can be approximatively predicted. Then, two different resonators have been studied, having edge lengths of 6 mm and 4 mm, respectively, to excite the main mode at two different frequencies, corresponding, in the present case, to values around 20 GHz and 27 GHz.

The experimental characterization of the proposed configurations was performed using an on-wafer system with coplanar probes, namely ground-signal-ground (GSG) tips touching the input/output short transmission lines used to feed the resonators. The central conductor of the probe brings the signal to the central conductor of the CPW, while the lateral tips are utilized as a ground reference, touching the lateral grounds of the CPW. A vector network analyzer (VNA), i.e., a measurement system commonly used in microwave engineering to characterize a multi-port device, has been used for recording the amplitude and phase of the reflected and transmitted signals. In our case, an HP8510C VNA (from Keysight Technologies, formerly Hewlett and Packard) is linked to a PM5 Karl Suss manual probe station with micrometric control for positioning the CPW probes. A PC is connected to remotely control the instruments. A simple schematic of the experiment is shown in Figure 7, detailing the measurement procedure. An essential step in the characterization is the calibration of the measurement system, intended to utilize the hardware and software necessary for de-embedding the measurement itself. In this case, a through-reflect-line (TRL) calibration was performed using standard transmission lines with different lengths (and phase shifts) determined by the line lengths. They are standard lines, designed in agreement with the frequency range imposed for the measurement, following design criteria now well-established in the scientific literature [32]. The calibration allows the artificial movement of the measurement plane at the device input/output, de-embedding the contribution of discontinuities and cables up to the device under test (DUT). In our measurements, the reference plane is given by the position of the probes shown in Figure 7. This measurement protocol, i.e., choice of probes, calibration, and measurements once the frequency range and resolution, and the input power, are chosen, is the most reliable technique for characterizing devices on-wafer. No limitations are encountered when properly designed CPW lines are used for the TRL calibration and one of the standard lines is verified after that, used as a test to validate the calibration itself.

In Figure 8, the measured frequency response is shown, with resonance frequencies *F_r_* working around 20 and 27 GHz, respectively. The *F_r_* values have been chosen at the design level to be coherent with current satellite communications, focusing on the K-Band within the range currently considered for uplink and downlink operations. Then, the main resonance mode has been selected according to the chosen length for the triangle edge. Since two different frequencies have been selected, despite the presence of high order modes, focus has been given to the main resonance and to its modifications induced by the internal complexity of the triangle.

First, the increase in the internal complexity of the triangle contributes to a shift in the resonance frequency.

Unbalanced configurations are obtained if no attention is paid to balancing the RH and LH contributions of the structure, which would not be ideal and would always be affected by parasitic contributions. For this reason, close to the resonance frequency, a gap is opened in the dispersion relation [10]. It is also true that when the wavelength is much larger than the discontinuities encountered during the propagation, the structure can easily be considered balanced. During the passage from RH to LH behavior, a TL section will be characterized by a dispersion relation passing from negative to positive values of the excited wavevector [10]. Then, it would be natural to expect that at resonance, a resonating structure could pass from an ordinary response to a metamaterial one, with a change in the sign for the dielectric constant and the magnetic permeability. Since we are focused on resonator configurations, this response should be concerned with a narrow-band behavior limited to the resonance frequency and its boundaries. Using the approach of the equivalent circuit proposed in this contribution, the impedance values have been plotted around the resonance frequency for different configurations and frequencies of operation, comparing C0 and C1.

Figure 9 plots the imaginary parts of *Z*_1_ and *Z*_2_′ for the 20 GHz resonator in the C0 and C1 configurations. The *Z*-values have been obtained from the *S*-parameters using literature formulas, as specified before in the text. At resonance and surrounding frequencies, C0 always shows a monotone response of the two quantities, which do not change when the sign passes through the frequency of resonance at 20.16 GHz, while, in the case of C1, the right side of the resonance peak exhibits negative values of both ${Im(Z}_{1})$ and $Im\left(Z_{2}^{\prime }\right)$ up to frequencies of a few hundred MHz from the resonance frequency. A possible interpretation of this behavior returns to the generally expected response in Figure 3, i.e., around the resonance frequency, the C0 configuration, despite nominally not being a fractal configuration, always exhibits a metamaterial response because $Im\left(Z_{2}^{\prime }\right)>0$ and ${Im(Z}_{1})<0$ over the entire resonance frequency band. The C1 configuration exhibits a more complicated response because we can distinguish three frequency ranges: (1) before the resonance, we have $Im\left(Z_{2}^{\prime }\right)>0$ and ${Im(Z}_{1})<0$, nominally corresponding to a metamaterial structure, (2) just after the resonance, both quantities are negative, thus giving evidence for a dominant capacitive contribution on both branches of the equivalent T-circuit, and finally, (3) the response is inverted, with $Im\left(Z_{2}^{\prime }\right)<0$ and ${Im(Z}_{1})>0$ at the end of the resonance curve, corresponding to the response of an ordinary TL.

Figure 10 shows a result similar to the previous one, with a metamaterial response for C0 and mainly a capacitive contribution for C1 around the resonance frequency.

A more complicated configuration has been studied, including three coupled resonators with the C0 and C1 configurations, also, in this case, mirrored to the central conductor of the CPW feeding line and for the same frequency ranges, i.e., close to 20 GHz and 27 GHz. The complete structure of the coupled resonators looks like a hexagon because of the mirrored arrangement. It might be stressed that a symmetric configuration was always the best solution for all the studied resonators from the electrical performance point of view, mainly because a CPW favors a symmetric coupling considering the specific geometry of the feeding line [31,32,33].

The layout of the hexagonal resonators implemented by the C0 and C1 configurations is shown in Figure 11, while in Figure 12, the Transmission Parameter S_21_ is plotted.

The results in Figure 12 are not easy to understand, as only the C1 configuration is critically matched for both frequency ranges, exhibiting a deep notch. For the C0 triangles, the resonance is unexpectedly strongly depressed around 20 GHz, while additional modes around 21.5 GHz are excited for both structures based on the C0 and C1 resonators. This result is possible when exciting resonating modes with an external coupling. A modal analysis can help determine the position of the maximum of the fundamental mode, even if, for practical applications, a simple configuration is encouraged and the principal mode cannot always be excited with the chosen feeding system. At 27 GHz, the resonance of C1 is well represented, but C0 has a much lower intensity. The above findings prove that a general criterion to match the single resonator and the coupled ones is not straightforward. Fixing some geometrical parameters, like the distance between coupled resonators and their distance to the central conductor of the CPW feeding line or the length of the input/output feeding lines, cannot contribute to shifting the operative frequency and simultaneously preserving the exact electrical matching when increasing the internal complexity. Despite the wideband response of the CPW alone, the coupling between CPW and resonators and the mutual coupling among the resonators needs further refinement. The distance between the individual triangles in the C0 configurations should be optimized, even if C0 performs well as a single resonator. At the same time, C1 is suitable for single and coupled structures, exhibiting a good response in both cases. The results for C0 and C1 hexagons are plotted in Figure 13 and Figure 14.

The previous figures show that, despite showing different levels of electrical matching, the C0-based structures can be reasonably considered metamaterial-like around the resonance frequency. In contrast, C1 structures are better matched, but a different comment is necessary due to the response of the equivalent $Im\left(Z_{2}^{\prime }\right)$. In this case, a configuration with C1 arranged in a hexagon with a near-to-zero value has been obtained for both frequencies at resonance while ${Im(Z}_{1})<0$ in the same range. Then, the resonance frequency can delimit two regions close to the resonance *F_r_*: (1) for *F* < *F_r_*, the structure shows a metamaterial response with $Im\left(Z_{2}^{\prime }\right)>0$ and ${Im(Z}_{1})<0$, and (2) for *F* > *F_r_*, and the two imaginary values are negative, with evidence for a purely capacitive response.

The above considerations are a confirmation that a nominal fractal or metamaterial-inspired structure needs to be examined in detail, including the entire geometry, to confirm the possible metamaterial response because other details, like the feeding line, the boundaries, and the coupling mechanisms, in the case of multiple resonators, are essential components of the entire framework. In particular, when many resonators are coupled together, the importance of the feeding line decreases, leading to a vanishing $Im\left(Z_{2}^{\prime }\right)$, like in the case of an ordinary resonating structure.

## 5. Conclusions

This paper developed a simple approach to experimentally derive the equivalent circuit of planar microwave components, namely microwave resonators fed by CPW lines, to determine the ordinary or metamaterial response of the device. The method is based on *Z*-parameters analysis and specifically on the sign of $Im\left(Z_{2}^{\prime }\right)$ and ${Im(Z}_{1})$, i.e., the two branches of the T-model for a transmission line section, including the resonator and the feeding line, respectively. The T-model has been used and related to the experimental data to derive the possible metamaterial response of the investigated structures, analyzing the sign (positive or negative) of the impedances defined along the branches of the T-network. In particular, the studied configurations were triangular planar resonators coupled with a CPW line and symmetrically arranged to the central conductor of the CPW. According to the proposed method, the RH (ordinary) or LH (metamaterial) nature of the device was investigated, concluding that the possible LH response of the equivalent circuit confirms the device’s metamaterial nature, i.e., a series capacitor followed by a shunt inductor (LH structure) opposed to the ordinary series inductor with a shunt capacitor (RH configuration). Two triangular patches, designed with sizes fulfilling the requirement of 20 GHz and 27 GHz operation, have been compared with Sierpinski triangles with the same edge length of the entire patch, and with the first level of complexity, classically considered a fractal shape. Moreover, the same building blocks have been used to manufacture hexagonal structures, coupling three triangles and mirroring them with respect to the CPW line. The results obtained from the analysis of the inferred *Z*-parameters gave evidence for an equivalent circuit not always related to a metamaterial behavior. The single resonators and the coupled ones yield different responses despite the presumed metamaterial nature due to the fractal design for C1 and the ordinary one for C0. As a result, the complexity of the structure must include the analysis of the feeding system and the boundaries of the resonators, involving parasitic contributions not easily obtainable with ab initio calculations or de-embedded with a calibrated measurement, and is strictly related to the technology and to the necessity of having a feeding system. Then, the measured C0 configuration, i.e., the simple patch, appeared to be more clearly related to the metamaterial response despite the non-fractal shape. At the same time, the C1 structures (first-level complexity of the Sierpinski triangle) could be even better coupled from an electrical point of view but do not necessarily exhibit a metamaterial response. This result is particularly evident in the coupled resonators, where the boundary conditions are less critical with respect to the dominant area occupied by the entire resonating structure, thus behaving as a simple resonator, with $Im\left(Z_{2}^{\prime }\right)=0$. The above findings show that even a metamaterial-inspired resonating structure could behave as an ordinary material or vice-versa, depending on the coupling mechanisms and additional equivalent lumped components to be considered.

In conclusion, the approach proposed in this contribution aims to determine experimentally the equivalent dielectric constant and magnetic permeability of a metamaterial-inspired structure or possibly for configurations not necessarily considered from the beginning as metamaterial resonators. Still, they can be recognized as left-handed (LH) structures if considered as a whole. This is particularly important when not only single resonators are designed, but also arrays are considered, typically accompanied by complicated coupling solutions and feeding networks. Further efforts are also needed to reconstruct the spectrum of the individual resonators, as exhaustive work has only been conducted for the microstrip excitation for antennas and, in part, for resonators, but coplanar waveguides need further work to understand in detail the spectrum excited in a coplanar configuration.

## Figures and Tables

**Figure 1 sensors-24-02212-f001:**
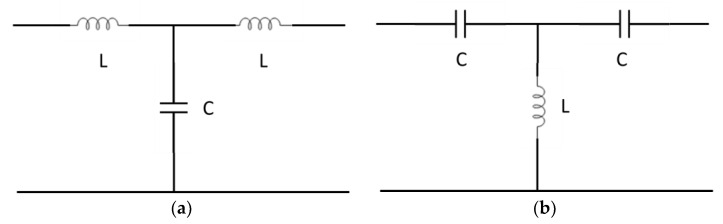
Transmission line equivalent circuit for (**a**) classical configuration and (**b**) metamaterial structure. L and C are the inductance and capacitance respectively.

**Figure 2 sensors-24-02212-f002:**
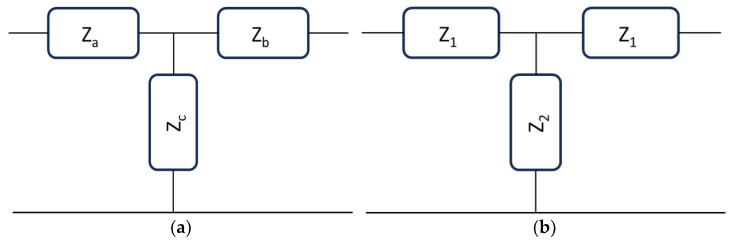
General TL described: (**a**) through arbitrary impedances and (**b**) using a symmetric and reciprocal cell, with equal impedances at the input and output ports.

**Figure 3 sensors-24-02212-f003:**
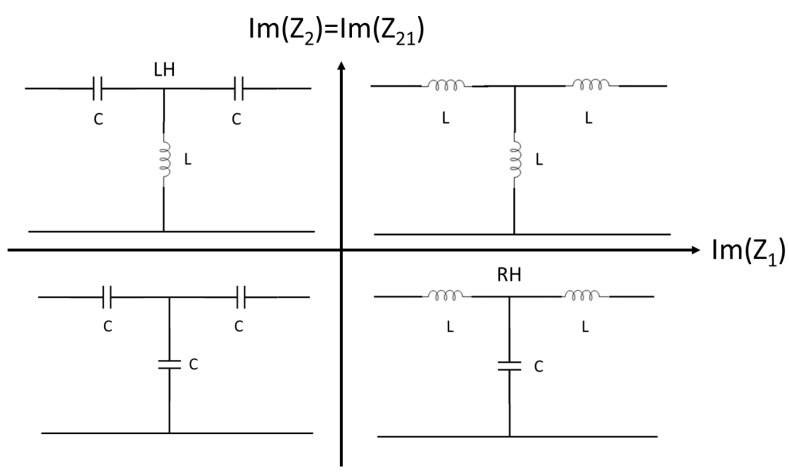
Equivalent elementary cells of an ideal TL, depending on the sign of the reflection and transmission impedances.

**Figure 4 sensors-24-02212-f004:**
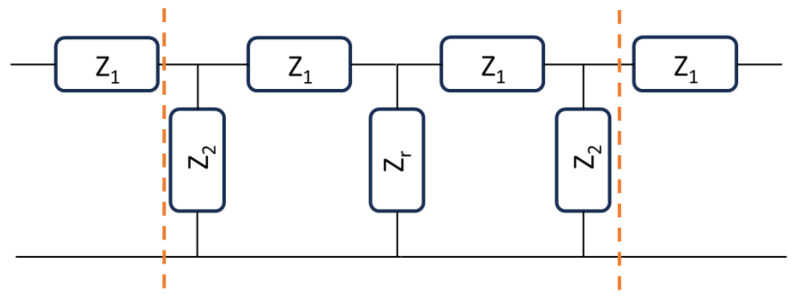
Equivalent circuit for a resonator fed by two transmission lines.

**Figure 5 sensors-24-02212-f005:**
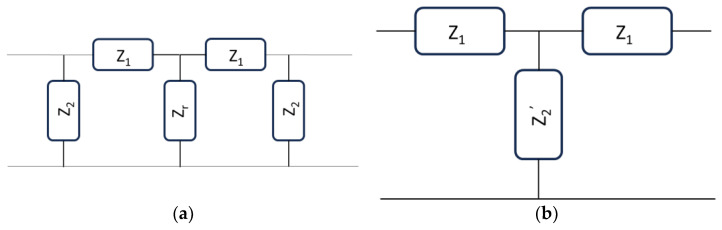
A Simplified approach to determine the potential metamaterial response of the resonator fed by two transmission lines. In (**a**), the central network of the entire structure is reproduced with all the components, while in (**b**), the simplified T-model with Z_2_′ substituting the intermediate network is proposed.

**Figure 6 sensors-24-02212-f006:**
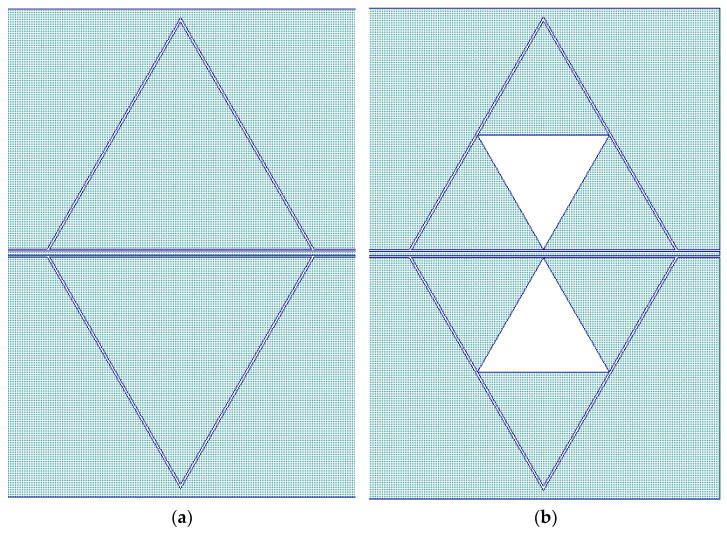
Triangular resonators: (**a**) full metal patch, and (**b**) Sierpinski triangle with first-level internal complexity. The shape is used for the 20 GHz nominal frequency of operation, with the triangle edge of 6 mm, while the same shape has been scaled down to a 4 mm for a higher frequency of resonance (close to 27 GHz).

**Figure 7 sensors-24-02212-f007:**
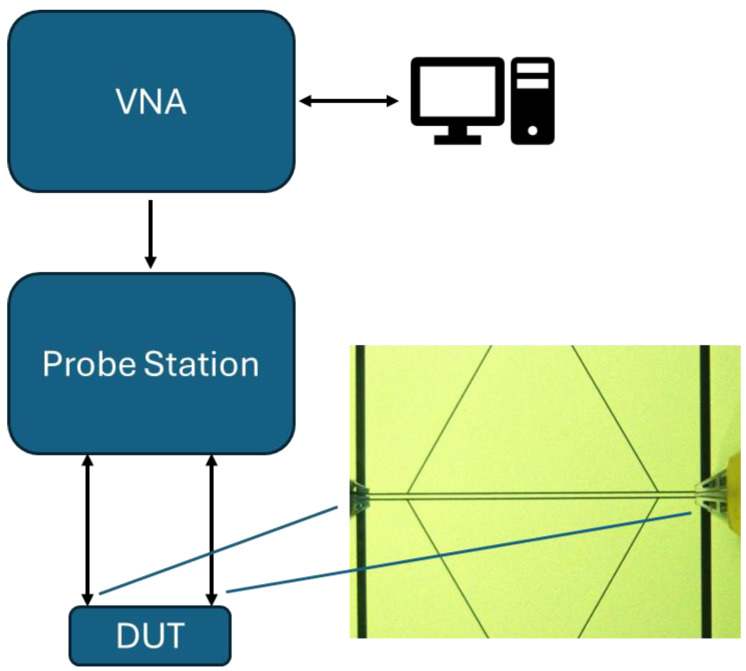
Schematic of the measurement system, composed of a VNA and a probe station remotely controlled by a PC. The device under test (DUT) is connected at the end of the cables by means of GSG coplanar probes, and the calibration is valid from the ends of the tips.

**Figure 8 sensors-24-02212-f008:**
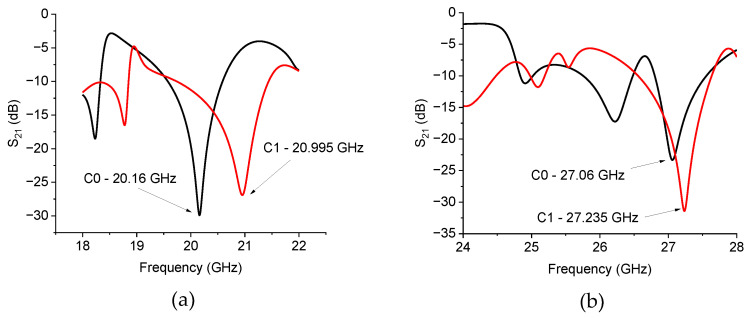
The frequency response for the triangles C0 and C1 for the two configurations: (**a**) 20 GHz and (**b**) 27 GHz.

**Figure 9 sensors-24-02212-f009:**
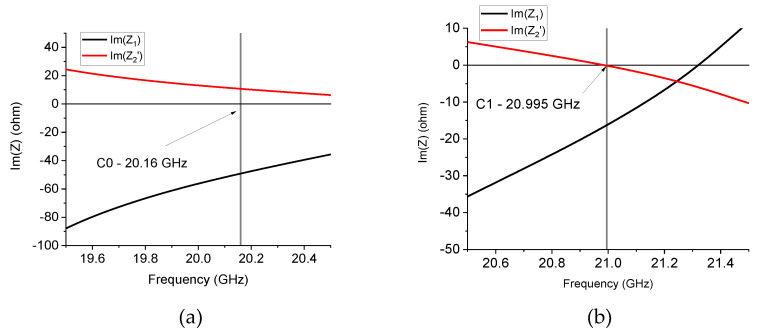
*Im*(*Z*_1_) and Im(*Z*_2_′) equivalent impedances for the C0 and C1 triangles working around 20 GHz. C0 always exhibits *Im*(*Z*_1_) < 0 and *Im*(*Z*_2_′) > 0 around the frequency of resonance. For C1, the two quantities are negative on the right side of the resonance frequency, and there is a transition below the resonance.

**Figure 10 sensors-24-02212-f010:**
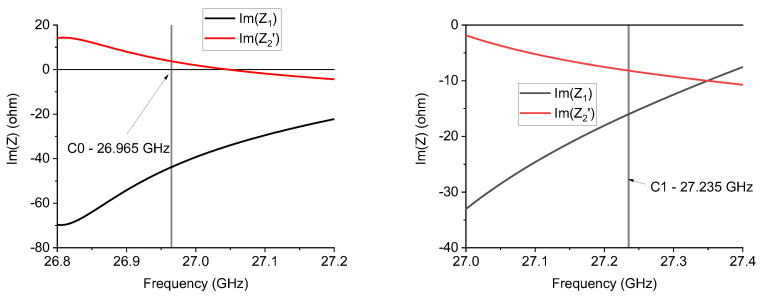
*Im*(*Z*_1_) and *Im*(*Z*_2_′) frequency response for the C0 and C1 triangles working close to 27 GHz.

**Figure 11 sensors-24-02212-f011:**
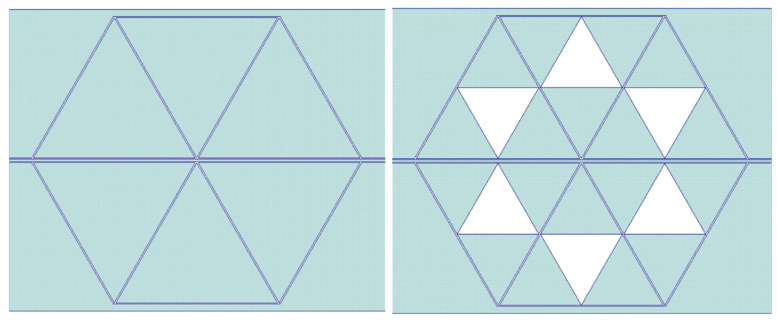
Triangular resonators are arranged in a hexagonal geometry by coupling three of them and mirroring the structure with respect to the central conductor of the CPW feeding line. The figure is for the 20 GHz resonators, and the 27 GHz configuration has been scaled down, as in the case of the individual triangles.

**Figure 12 sensors-24-02212-f012:**
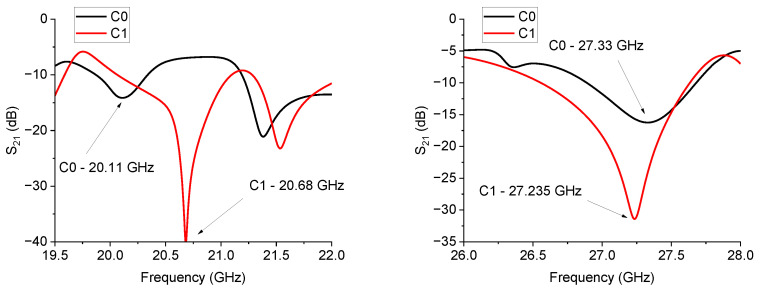
Resonance frequency response for the 20 and 27 GHz hexagonal structures.

**Figure 13 sensors-24-02212-f013:**
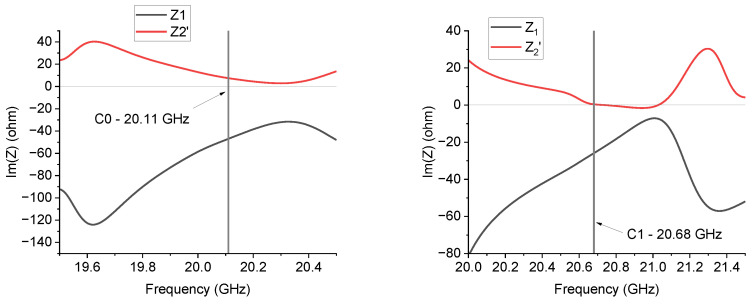
*Im*(*Z*) for the 20 GHz hexagonal resonators.

**Figure 14 sensors-24-02212-f014:**
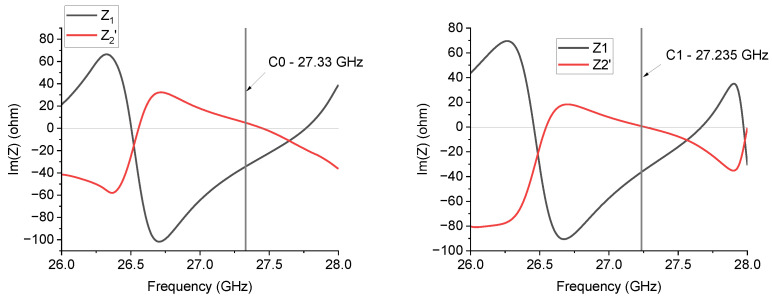
*Im*(*Z*) for the 27 GHz hexagonal resonators.

## Data Availability

Data availability is guaranteed upon requirement.

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
