# Peer review of "Equivalent Circuits for Microwave Metamaterial Planar Components"

_sensors, 2024, doi:10.3390/s24072212_

Round 1

Reviewer 1 Report

Comments and Suggestions for Authors

The paper presents an in-depth study of equivalent circuits for microwave propagation and resonance in metamaterial planar components. The focus on planar resonating devices and the calculation of effective values for the dielectric constant and magnetic permeability using impedance notation is commendable. However, there are several areas where the paper could benefit from further clarification and elaboration.

1.     This paper presents an important and timely contribution to the field of microwave metamaterial planar components. However, several key areas require significant revision and clarification to ensure the clarity, accuracy, and impact of the work.

2.     The use of technical terminology and notation can be inconsistent and confusing. For example, the term "MM" is used interchangeably with "metamaterial," which can be unclear for readers not familiar with the field. It is recommended to standardize the terminology throughout the paper.

3.     The paper briefly mentions Veselago's seminal theoretical paper, but fails to establish a strong theoretical foundation for its own work. It would be beneficial to delve deeper into the theoretical aspects of metamaterials and how they relate to the research presented in this paper.

Furthermore, the discussion on the dielectric constant and magnetic permeability lacks sufficient detail and context. Consider expanding on these concepts, their significance in the context of microwave metamaterial planar components, and their relationship to the equivalent circuits being studied.

4.     The paper lacks a detailed description of the methodology and experimental setup used to derive the equivalent circuits and calculate the effective values for the dielectric constant and magnetic permeability. It is crucial to provide a comprehensive description of the methods, including any assumptions, limitations, and potential biases.

Additionally, it would be helpful to include figures or diagrams that visually represent the experimental setup and the equivalent circuits, making the information easier to understand and follow.

5.     The analysis and discussion sections need to be significantly expanded and elaborated. The paper briefly mentions the comparison of equivalent circuits and the derivation of a lumped element modeling, but fails to provide a thorough analysis and interpretation of the results.

Consider discussing the implications of the results, comparing them to previous studies or theoretical predictions, and discussing any potential limitations or caveats.

6.     The conclusion section is relatively brief and lacks a clear summary of the main findings and their significance. It would be beneficial to provide a more comprehensive conclusion that highlights the key contributions of the paper, any limitations or future work that needs to be done, and the potential impact of the research on the field of microwave metamaterial planar components.

Overall, this paper has the potential to make a significant contribution to the field of microwave metamaterial planar components, but it requires significant revision and clarification to ensure its clarity, accuracy, and impact. Focusing on improving the theoretical foundation, methodology, and analysis will help to strengthen the paper and make its contributions more evident.

Author Response

Please, look at the attached file.

Reviewer 2 Report

Comments and Suggestions for Authors

The reviewer is sorry to have to express his deep misunderstanding of the manuscript. Perhaps his long-lasting knowledge of metamaterials is not sufficient facing the level of scientific content.

In the opinion’s reviewer, the following points must be clarified in the revised version.

In many places, authors speak about experiments. I do not see where measurements are presented. If the author means experiments using simulations, this should be clarified in the manuscript, as well as the software used for them.

Also, the goal of the work needs to be clarified. In the conclusion and other places, it is said that the equivalent circuit of the MM structures will be determined. This is not exact, since the equipment circuit is already fixed by the selection of figure 2, and 3. What is studied in the paper is the frequency behavior of the Z1 and Z2 elements of the equivalent circuit, to determine the LH or RH behavior. Please clarify in a revised version.

Also why are the results for both C0 and C1 structures presented at two different frequencies, 20 and 27 GHz? What happens between 23 and 24 GHz at figure 7, and why do the curves for CO and C1 show no continuity from Figure a  to Figure b?

It is difficult to understand the real benefits of the introduction of impedance Z2’ associated with access transmission lines. As far as the behavior of the MM solely is needed, well-known de-embedding procedures are usually applied to access directly the Z1 and Z2 elements of the corresponding resonator.

Finally, authors have to clarify/rewrite lines 256-260 which are difficult to understand.

Author Response

Please, look at the attached file.

Round 2

Reviewer 1 Report

Comments and Suggestions for Authors The authors have addressed all points in my review. I recommend accepting the paper for publication.

Reviewer 2 Report

Comments and Suggestions for Authors

answers to my comments are adequate, as well as changes made.